# Effect of Zirconium Doping on Electrical Properties of Aluminum Oxide Dielectric Layer by Spin Coating Method with Low Temperature Preparation

**Yue Zhou [1], Zhihao Liang [1], Rihui Yao [1,\*], Wencai Zuo [1], Shangxiong Zhou [1], Zhennan Zhu [1], Yiping Wang [2], Tian Qiu [3], Honglong Ning [1,\*] and Junbiao Peng [1]**

[1] State Key Laboratory of Luminescent Materials and Devices, Institute of Polymer Optoelectronic Materials and Devices, South China University of Technology, Guangzhou 510641, China; 201730341030@mail.scut.edu.cn (Y.Z.); 201530291443@mail.scut.edu.cn (Z.L.); 201730321407@mail.scut.edu.cn (W.Z.); 201820117973@mail.scut.edu.cn (S.Z.); mszhuzn@mail.scut.edu.cn (Z.Z.); psjbpeng@scut.edu.cn (J.P.)

[2] State Key Laboratory of Mechanical Structural Mechanics and Control, Nanjing University of Aeronautics and Astronautics, Nanjing 210016, China; yipingwang@nuaa.edu.cn

[3] Faculty of Intelligent Manufacturing, Wuyi University, Jiangmen 529000, China; timeqiu@hotmail.com

\* Correspondence: yaorihui@scut.edu.cn (R.Y.); ninghl@scut.edu.cn (H.N.)

**Abstract:** In recent years, significant efforts have been devoted to the research and development of spin-coated $Al_2O_3$ thin films, due to their large band gaps, high breakdown voltage and stability at high annealing temperature. However, as the alumina precursor has a large surface energy, substrates need to be treated by plasma before spin coating. Therefore, to avoid the expensive and process-complicated plasma treatment, we incorporated zirconium nitrate into the aluminum nitrate solution to decrease the surface energy of the precursor which improve the spreadability. Then, the electrical performances and the surface morphologies of the films were measured. For comparison, the pure $Al_2O_3$ films with plasma treatments were also prepared. As a result, after low temperature annealing (200 °C), the relative dielectric constant of $Zr–AlO_x$ spin-coated thin-film MIM (Metal-Insulator-Metal) devices can reach 12 and the leakage current density is not higher than $7.78 \times 10^{-8}$ A/cm$^2$ @ 1 MV/cm when the concentration of zirconium nitrate is 0.05 mol/L. The Aluminum oxide film prepared by zirconium doping has higher stability and better electrical properties than the pure films with plasma treatments and high performance can be attained under low-temperature annealing, which shows its potential application in printing and flexible electronic devices.

**Keywords:** zirconium doping; aluminum oxide; spin coating; low temperature annealing; dielectric layer

## 1. Introduction

Regarded as one of the keys for the stability of thin-film transistors, the dielectric layer has a great influence on the performance of thin film transistor (TFT) devices. With the development of wearable flexible devices, field effect transistors are facing the challenges of low-temperature fabrication, size reduction and less consumption [1–3]. However, as for traditional $SiO_2$ dielectric layers, the reduction in size will produce a large leakage current owing to the tunneling effect [4]. In recent years, metal oxide gate dielectric has been a hotspot due to its advantages of low driving voltage, large capacitance and high electrical stability [5–10]. Moreover, the oxide dielectrics are relevant to emerging thin-film transistor technologies, such as the ones based on organic semiconductors, amorphous metal-oxide semiconductors, semiconducting carbon nanotubes and two-dimensional semiconductors [11–15].

In the studies of metal oxide dielectric layers, $Al_2O_3$ has a high dielectric constant ($k$ = 8.6–10) and large band gap (~9 eV) [16]. The leakage current density is lower than that of the other gate materials with the same thickness [17–20]. Avis et al. found the TFT made with solution processed $AlO_x$ gate dielectric presents high field-effect mobility with lower interface trap density [21]. Ganesan et al. considered $Al_2O_3$ is a promising gate insulator due to its low leakage current density, higher dielectric constant and thermal stability [22]. As one of the key factors affecting the performance of TFTs, the dielectrics can be used to control the switching of thin film transistors through gate voltage. Several methods towards improving the performance and affordability of dielectrics are mentioned in the previous study including adopting polymer dielectrics [23] and patterning techniques [24]. The dielectric with a high dielectric constant can accumulate more charges, thus improving the conductivity of the semiconductor. Dielectric layers with smooth surface can reduce the scattering of carriers between the dielectric and the semiconductor. Therefore, the properties of the dielectric have a great effect on the performance of the devices. Most oxide dielectrics are fabricated by a vacuum process, such as plasma-enhanced chemical vapor deposition [25], atomic layer deposition [26] and magnetron sputtering [27], typically requiring high temperature and complex processing. Instead, solution processes are appealing alternatives which enable a low processing temperature and lead to low cost manufacturing, simplicity and high throughput of TFT arrays [28–30]. Nevertheless, the large surface energy of aluminum nitrate precursor solution makes the solution difficult to be applied into spin coating, leading to the problem of uniformity. To address this issue, plasma oxygen treatment (plasma) is usually used to pretreat the surface of the substrate [31]. However, the plasma treatment prolongs the process time and limits the type of substrate materials, so we intend to use the doping method to optimize the precursor solution. Among many candidate high-k materials, $ZrO_2$ has a high dielectric constant ($k$ = 25) and a large band gap (~7.8) [32,33]. In our former experiment, we have employed Zr-mixed $AlO_x$ films which can combine the properties of both [34]. During further study, we found that with the addition of zirconia, the $AlO_x$ films could be spread well without plasma. Therefore, here we adopted the method of Zr doping to achieve a better spreading. High-quality metal oxide dielectric often requires higher annealing temperature [35], whereas it contradicts the low-temperature processing required by flexible devices [36]. Previous research has reported that the doping of Zr can form oxide skeletons with fewer defects in relatively low temperature and lower processing temperature [37,38]. However, the perspectives of reducing the surface tension of the solution and optimizing the spinning process by doping zirconia have not been studied. Here we study from the aspect of process optimization and focus on comparing the difference between doping and plasma.

In this study, we added zirconium nitrate with different concentrations into aluminum nitrate solution to overcome the problem of wetting and tested the effect of different annealing temperature. The results show that Zr doping not only can increase the wetting ability and smooth the surface of aluminum oxide dielectric layer, but also can increase the uniformity and density of aluminum oxide dielectric layer. The MIM devices of Zr–$AlO_x$ fabricated by spin-coating show high relative dielectric constant and low leakage current density at a low annealing temperature process. It also proves the applicability of this approach for printed and flexible electronic devices.

## 2. Material and Methods

For the deposition of $Al_2O_3$ and Zr–$AlO_x$ gate dielectric layers by a solution process, precursor solutions for each gate dielectric were prepared by dissolving metallic precursors in an ethylene glycol monomethyl ether (EGME) solvent. For the $Al_2O_3$ gate dielectric, aluminum nitrate nonahydrate ($Al(NO_3)_3 \cdot 9H_2O$) was dissolved in EGME with a concentration of 0.3 M. For the Zr–$AlO_x$ gate dielectrics, aluminum nitrate nonahydrate and zirconium nitrate pentahydrate ($Zr(NO_3)_4 \cdot 5H_2O$) were dissolved in EGME (Al: 0.3 M, Zr: 0.01–0.1 M). After stirring at room temperature for 24 h, the clarified precursor solution was obtained. The solutions were aged under ambient conditions for 24 h and their surface tension were then measured by an Attension Theta Lite (TL200, Biolin Scientific, Gothenburg, Sweden). According to the spin coating solvent volatilization model proposed

by Meyerhofer, the relation between film thickness, $h$ and spin rate, $\omega$, obeys the power law relation of the form $h \propto \omega^{-\frac{2}{3}}$ [39]. The thickness is positively correlated with the concentration of the precursor used for spin-coating [40]. Since lower spin speeds lead to rougher surfaces and thicker films, each solution was spin-coated on bare glass and ITO glass (5000 rpm, 30 s). After deposition, the films were preheated at 150 °C for 10 min to form solid films by evaporating the solvent. The two process steps, spin coating and subsequent hot plate annealing, were repeated three times in order to have a suitable thickness of the layer. The films were then thermally annealed at 150, 200 and 250 °C, respectively for 1.5 h on a hotplate in air ambient. Moreover, $AlO_x$ solutions follow the same steps on the plasma treatment glass substrate before spin coating. The average thickness of the spin-coated thin ion gels were about 100 nm. The structure of Zr–$AlO_x$ films were analyzed by X-ray diffraction (XRD, EMPYREAN, PANalytical, Almelo, The Netherlands) and the chemical characteristics of films were investigated by Fourier Transform Infrared (FT-IR, ATR Accessory, Nexus). The surface morphology and the roughness of the films were observed by using (BY3000) atomic force microscope (AFM, Being Nano-Instruments, Beijing, China). Film thicknesses are measured by probe surface profiler (model: Dektak 150, Veeco, Plainview, NY, USA). For the evaluation of the dielectric properties of $AlO_x$ and Zr–$AlO_x$ films, an 80-nm-thick silver electrode was deposited on the gate dielectrics to obtain MIM devices (ITO/$AlO_x$/Ag and ITO/Zr–$AlO_x$/Ag). The capacitance–voltage characteristics and current–voltage characteristics of MIM devices were carried out by using Keithley4200 (Tektronix, Beaverton, OR, USA).

## 3. Results and Discussion

### 3.1. Surface Tension of Precursor Solution

To understand how the Zr doping affected the surface energy of alumina precursor, the surface tensions of the solutions were measured. Figure 1 shows the dropping images of solutions. The surface tension was calculated by the volume of the drops and concluded in Figure 1d. As depicted in Figure 1d, zirconia doping can reduce the surface tension of the aluminum nitrate solution. In general, the larger surface tension leads to a bigger contact angle with the same substrate. In addition, viscosity is related to the surface tension which plays a vital role on the spreadability of the film. Silverman and Roseveare once put forward an empirical equation as Equation (1), where $\eta$ is the viscosity, $\gamma$ is the surface tension, $A$ and $B$ are empirical constants for a given substance [41]. Sanyal and Mitra also postulated the following relations as Equation (2), where $\alpha$ and $b$ are empirical constants [42].

$$\gamma^{-\frac{1}{4}} = \frac{A}{\eta} + B \tag{1}$$

$$T \log \eta - \alpha \gamma = b \tag{2}$$

These equations indicate that the surface tension and viscosity of the liquid gas surface are positively correlated. On the basis of these theories and the reduction of solution's surface tension, we put forward reasonable hypotheses that the viscosity will reduce as the surface tension decreases. This may be attributed to the fact that doping zirconia can reduce the inter molecular forces in the solution, which reduces the resistance to liquid flow and results in a reduction in viscosity. Since a lower viscosity contributes to better spreadability and uniformity of the films, we can use the Zr-doping method to improve the surface morphologies. It can maintain the excellent electrical properties of alumina to meet the needs of spin coating without plasma treatment.

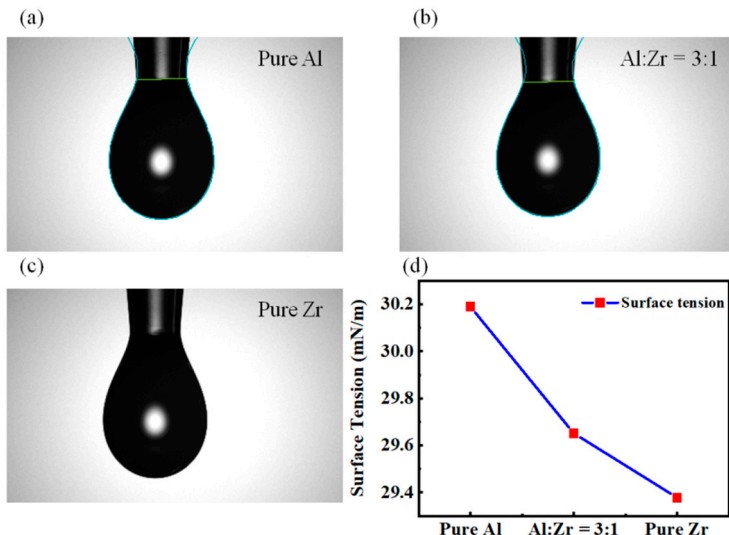

**Figure 1.** Dropping images. (**a**) Pure aluminum solution; (**b**) mixed zirconia and alumina solution (1:3); (**c**) pure zirconium nitrate solution; (**d**) surface tension of these solutions.

### 3.2. Surface and Structure of Zr–AlO$_x$ Films

The structural properties of Zr–AlO$_x$ films were characterized by XRD. The XRD spectra are shown in Figure 2. From Figure 2, no diffraction peaks are observed, and all the characterized films are practically in a non-crystalline phase. Since electric properties are vulnerable to structural characteristics, such as grain boundaries on surface, the amorphous state can effectively reduce the leakage current densities of the films [43]. In addition, the amorphous state of high-*k* gate dielectrics is also a desirable state because of their low surface roughness [4].

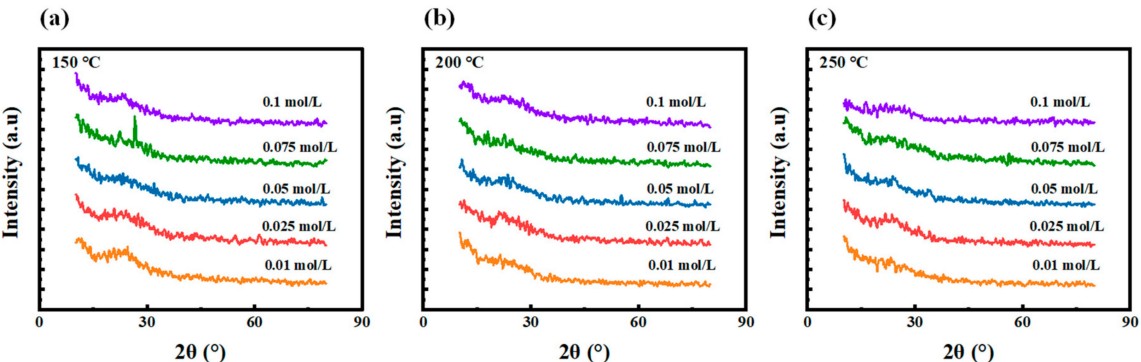

**Figure 2.** X-ray diffraction (XRD) spectra of Zr–AlO$_x$ films with different Zr concentrations: (**a**) annealing at 150 °C, (**b**) annealing at 200 °C, and (**c**) annealing at 250 °C.

In order to check the presence of the organic additive in the membrane, FT-IR spectroscopy was used. Figure 3 presents the FT-IR spectra of the films. The O−H stretching vibration shows a broad peak in the range 3000–3500 cm$^{-1}$ and a peak in 1500–1700 cm$^{-1}$ [44]. The peaks in the range of 1200–1700 cm$^{-1}$ correspond to nitrate deformation vibrations [45]. The carbonate species of solvent associated peaks are assigned to 700–1350 cm$^{-1}$. At different conditions of temperature and concentration, the chemical characteristics of membrane changed little.

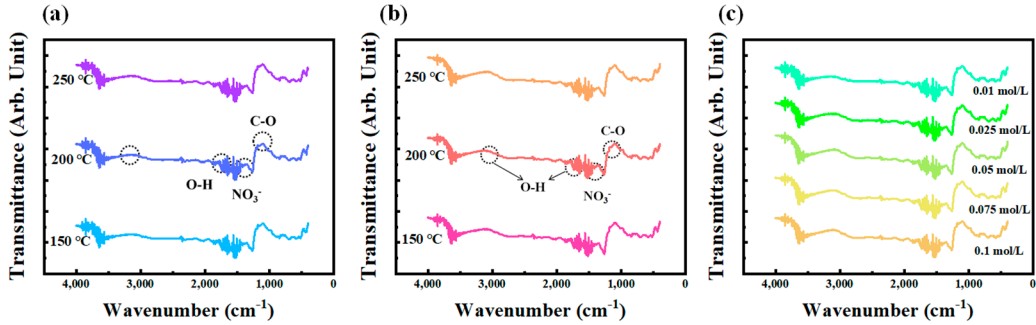

**Figure 3.** (**a**) films with plasma treatment; (**b**) Zr–AlO$_x$ films with 0.05-mol/L concentration under different annealing temperatures; (**c**) Zr–AlO$_x$ films annealed under 250 °C with different doping concentrations.

We used atomic force microscopy (AFM) to characterize the microscopic morphologies of films with the scanning area of 5 μm × 5 μm. The surface morphology of the film is described by root mean square roughness ($S_q$). The definition of $S_q$ is the standard deviation of the surface height of a specified area. According to the corresponding surface height analysis histograms as shown in Figure 4, the root mean square roughness ($S_q$) is calculated by the Equation (3) [46].

$$S_q = \sqrt{\frac{\sum_{i=1}^{n}(Z_i - Z_{avg})^2}{N}} \tag{3}$$

where $N$ is the number of surface points, $Z_i$ is the height at a point on the surface, $Z_{avg}$ is the average of $Z_i$.

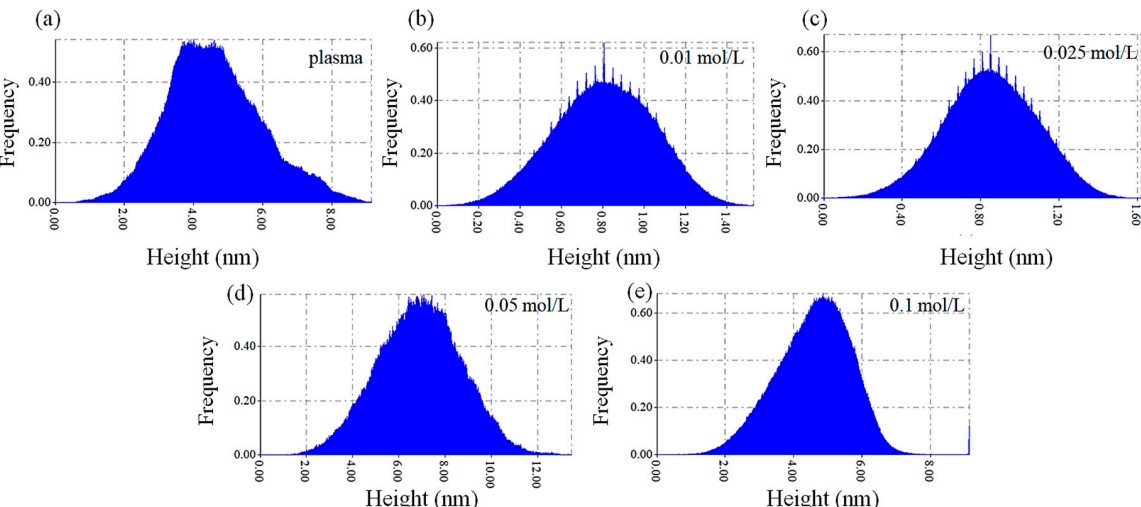

**Figure 4.** Corresponding surface height analysis histogram of films with different treatments under 200 °C, which is obtained by analyzing AFM images with software: (**a**) Pure aluminum solution with plasma treatment; (**b**) 0.01-mol/L Zr-doping solution; (**c**) 0.025-mol/L Zr-doping solution; (**d**) 0.05-mol/L Zr-doping solution; (**e**) 0.1-mol/L Zr-doping solution.

From Figure 4 the center height of pure AlO$_x$ film is found to be four nanometers, whereas with low Zr incorporations the surfaces are very uniform with a height mainly in 0.8 nm. As the Zr concentration increases to 0.05 mol/L, the height value increases with a more concentrated range, which leads to increase of $S_q$. However, it is observed that when the Zr concentration rises to 0.1 mol/L, the height becomes smaller. To better comprehend this phenomenon, we combine with the AFM images and go through a further analysis. Figure 5a–e shows AFM images of different films after 200 °C annealing process. At a low doping concentration, Zr can improve the surface morphology of films and achieve smooth surface (rms = 0.24 at 0.01 and 0.025 mol/L). With the Zr-incorporation

of 0.05 mol/L, the surface roughness increased to 1.84 nm (near to the level of plasma pretreatment film). This is probably because the short bond and the large bond energy of Zr–O result a distorted structure. However, an applicable high doping concentration can result in an even distribution and lower the $S_q$ value at 0.01 mol/L. As we know, a smooth surface is one of the necessary conditions for high-performance insulating materials. These are quite smoother surfaces compared to other materials, which provides its application in printed devices potential possibilities [47–49]. Moreover, the rms values of films under different annealing temperatures and doping concentrations are depicted in contour map (Figure 5f). Figures 4 and 5 show that the plasma-treated film is mostly rougher than other zirconium-doped films and its surface roughness (root mean square, RMS) reaches 1.39 nm. Therefore, the dielectric with doping Zr has potential to achieve better performances.

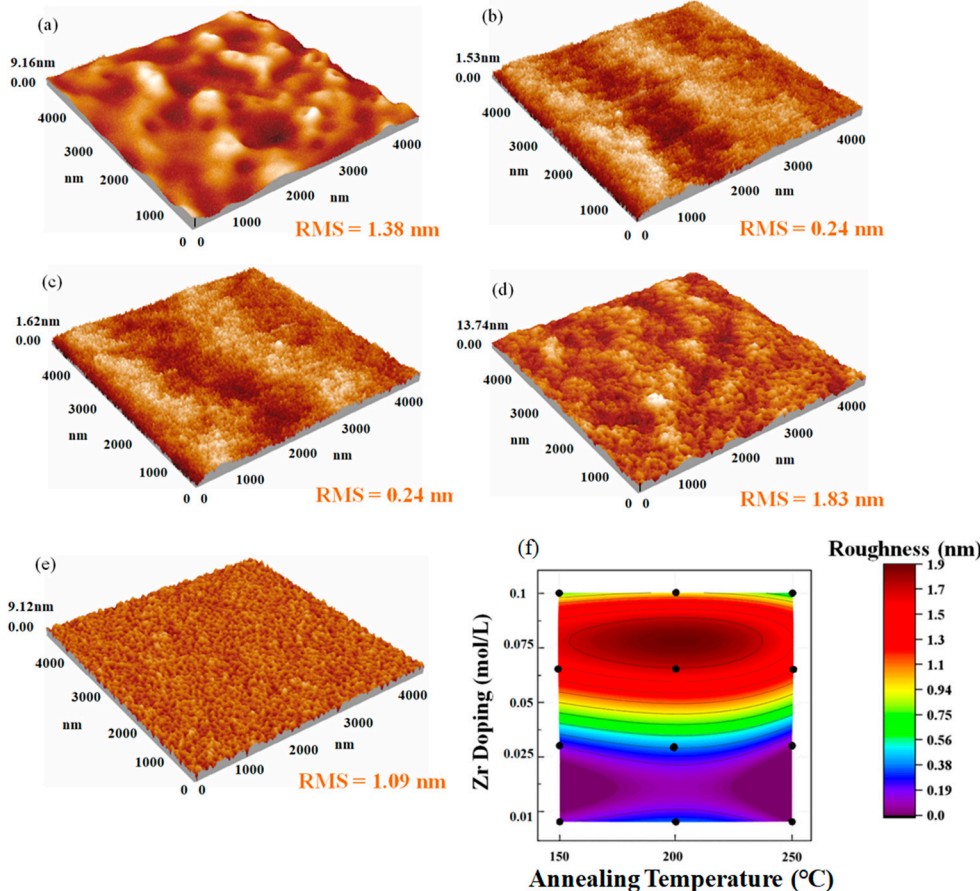

**Figure 5.** Atomic force microscope (AFM) images of different Zr–AlO$_x$ films at 200 °C annealing temperature. (**a**) Pure aluminum solution with plasma treatment; (**b**) 0.01-mol/L Zr-doping solution; (**c**) 0.025-mol/L Zr-doping solution; (**d**) 0.05-mol/L Zr-doping solution; (**e**) 0.1-mol/L Zr-doping solution; (**f**) Contour map of the relationship between annealing temperature and Zr-doping concentration of Zr–AlO$_x$ films surface roughness.

### 3.3. Electrical Characteristics of Zr–AlO$_x$ Films

We have measured the electrical characteristics of Zr–AlO$_x$ films based on the MIM device. MIM devices were fabricated using silver as electrode and its sketch map is shown in Figure 6. The devices were characterized by capacitance–voltage (*C–V*) and leakage current density-voltage (*I–V*) curves.

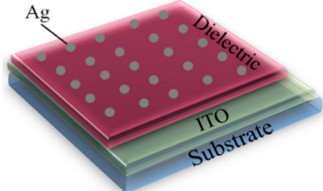

**Figure 6.** MIM (metal-insulator-metal) sketch map of Zr–AlO$_x$ films.

The dielectric characteristics of the MIM capacitor were measured by the Agilent B1500A (Keysight Technologies, Santa Rosa, California, CA, USA) in an ambient air environment. The CV maps were acquired under a DC voltage mode with the frequency of 1 MHz and QSCV Voltage of 0.1 V. In order to avoid inaccuracy brought by measurement and accidental phenomenon, we prepared two samples for each condition to measure their I–V characteristics. For each MIM device we choose three electrodes to repeat the test. The results of each type show consistent patterns.

The relative dielectric constants of the dielectric layers were calculated using capacitance and were concluded in Figure 7d. Figure 7 shows the relationship between the relative dielectric constant, Zr-doping concentration and the annealing temperature, Figure 7a–c are the capacitance values of the devices under 150, 200 and 250 °C. When the concentration of Zr doping is 0.05 mol/L, the spin-coated Zr–AlO$_x$ films exhibit good electrical characteristics with relative dielectric constants of 13.27 at 150 °C annealing, 11.96 at 200 °C annealing and 10.7 at 250 °C annealing. Correspondingly, we tested the electrical characteristics of the undoped solution spin-coated thin films at the same annealing temperature by plasma treatment and the relative dielectric constants were around 7 in the range of 150 to 250 °C. According to the measurement of relative dielectric constants, the Zr doping can enhance the electronic storage capacity of the films, probably attributed to the higher $k$ of zirconia itself.

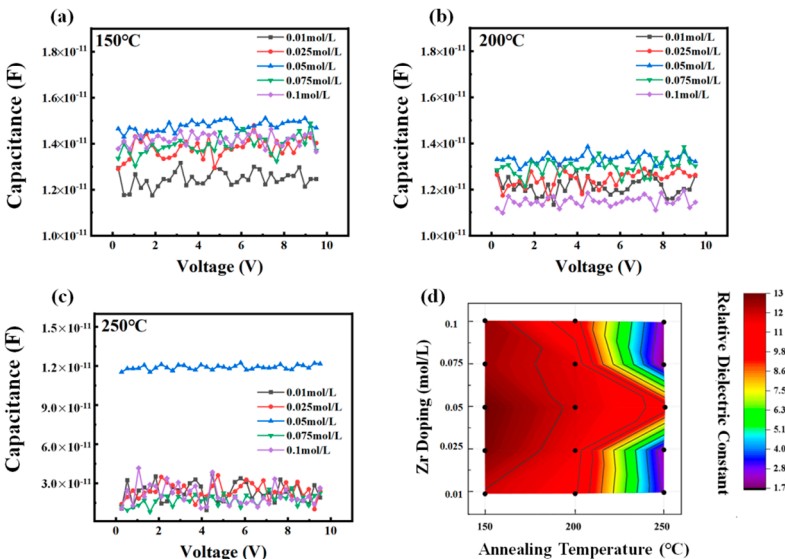

**Figure 7.** (**a**) The capacitance of films at 150 °C annealing; (**b**) capacitance of films at 200 °C annealing; (**c**) The capacitance of films at 250 °C annealing; (**d**) contour map plot of films' dielectric constants under different annealing temperatures and Zr-doping concentrations.

Figure 8 shows the main effects of relative dielectric constant carried by minitab data processing analysis. From Figure 8, it is found that the annealing temperature of the film is more significant than the Zr-doping concentration on the effect of the relative dielectric constants. With the increase of annealing temperature, the relative dielectric constant of Zr–AlO$_x$ film decreases. This is because when the spin-coated Zr–AlO$_x$ layer was annealed at a lower temperature, the dielectrics contained a smaller amount of hydroxide and nitrate ions. As the remaining NO$_3^-$ and OH$^-$ ions induced water, the electrostatically absorbed water molecules form electrical double layers in the Zr–AlO$_x$

film conducing to high capacitance performance [44,45,50]. Moreover, at about 250 °C, for the dehydroxylation reactions, the dielectric constant exhibits a rapid reduction [47].

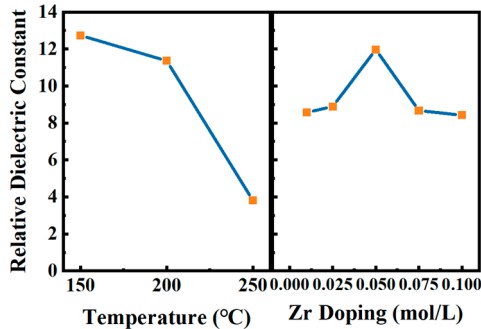

**Figure 8.** Main effects plot of Zr–AlO$_x$ relative dielectric constant.

In addition, as the Zr-doping concentration increases, the relative dielectric constant achieves the highest value at 0.05 mol/L. The higher constants probably originate from the uniform spreading, weak bond reduction and defect getting filled when moderate concentration is applied. The optimized process parameters are of great significance for obtaining high relative dielectric constant. The film under annealing temperature of 150 °C and a moderate Zr-doping concentration (0.05 mol/L) performs a highest relative dielectric constant.

Figure 9a–c shows the relationship between the leakage current density, Zr-doping concentration and annealing temperature, while the data of alumina film after plasma treatment was also added. Through calculating the differential resistance ($R_{diff} = dU/dI$) of MIM devices, we determine the shunt resistance ($R_{shunt} = R_{diff} (0 \text{ V})$) [51]. As shown in Figure 9d, the film with 0.05-mol/L Zr and 200 °C annealing achieves an ideal $R_{shunt}$ (7.36 × 10$^6$ Ω), whereas the films treated by plasma perform lower values (2.55 × 10$^6$ Ω at 150 °C, 1.16 × 10$^6$ Ω at 200 °C, 2.74 × 10$^6$ Ω at 250 °C). High shunt resistances can better the performance in dielectrics and reduce the leakage current. Compared with the films with Zr doping, the AlO$_x$ films prepared by plasma show higher leakage current densities (with the average of 1.32 × 10$^{-7}$ A/cm$^2$ @ 1 MV/cm). Under the Zr-doping concentration of 0.05 mol/L, the value of leakage current density keeps low and stable (as low as 6.92 × 10$^{-8}$ A/cm$^2$ @ 1 MV/cm at 200 °C) while in the case of other concentrations, the leakage present a large value at 150 and 250 °C.

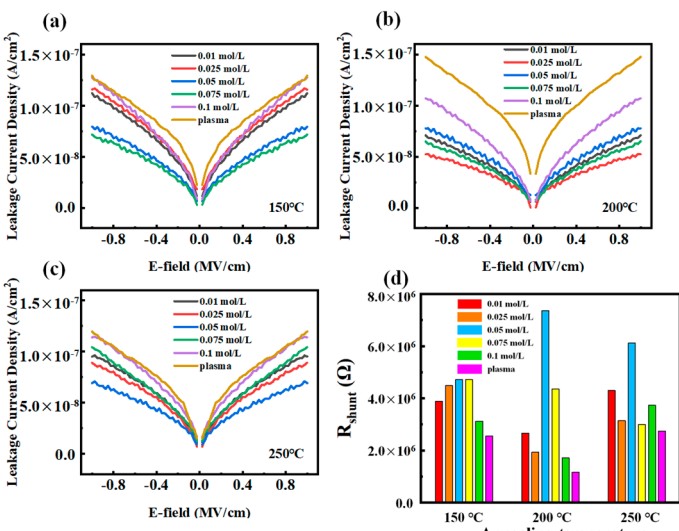

**Figure 9.** Zr–AlO$_x$ film leakage current at different annealing temperature: (**a**) 150 °C, (**b**) 200 °C, (**c**) 250 °C; (**d**) shunt resistance of films aforementioned.

Figure 10 shows the main effects of Zr–AlO$_x$ film leakage current densities. From Figures 9 and 10, it is found that the leakage current density first falls and then grows with a lowest leakage current density at 200 °C. In the range of 150 to 200 °C, the trapped charges and internal defects reduce at a higher temperature [52]. Moreover, a better evaporation of the organic residue can be conducive to reducing the leakage current. However, when the annealing temperature rises to 250 °C, the more complete oxidation of the zirconium results in more pronounced differences in bond energies and bond lengths between the zirconia and alumina, resulting in more oxygen vacancies that can be plugged into these displaced distances. Attribute to the facts above, the films under 200 °C annealing perform low leakage current densities. Since the low temperature will lead to a reduction of the electrical properties of the films owing to the organic residue, doping zirconia can achieve high performance at low temperatures to provide potential for low-temperature processing.

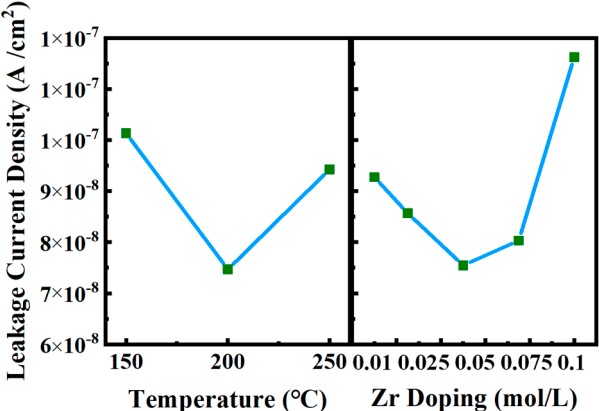

**Figure 10.** Main effects plot of Zr–AlO$_x$ leakage current density.

Doping Zr improves the electrical properties of the films by replacing weak oxygen bonds in AlO$_x$ (Al–O (511 kJ·mol$^{-1}$), Zr–O (766.1 kJ·mol$^{-1}$)) [53,54]. For the alumina films, the weak bonds to oxygen construct a loose oxide structure [55]. Thanks to the presence of Zr atoms, the larger atomic size (Ion radius Zr$^{4+}$:Al$^{3+}$ = 80:50) leads to the decrease of ion conduction and makes the Zr–AlO$_x$ films more densified [19,34]. The addition of high-bond-energy zirconium replaces aluminum to form a more compact and stable structure, resulting in improved performances of the dielectrics.

However, as for the effect of concentration, the leakage current density increases and then falls with the Zr incorporation. This may be owing to the larger relative atomic mass (Zr = 92, Al = 27), Zr mostly tend to sink in the gel during the evaporation of solution and finally deposit at the bottom of the dielectric layers. As more Zr incorporate, zirconium atoms will attract more oxygen, adding oxygen vacancies on the surface which adversely affect electrical properties. This trend is reflected in both the leakage current density and capacitance. Therefore, the optimized film preparation process conditions: 0.05-mol/L Zr doping and annealing temperature between 150 and 200 °C. Under these conditions, its relative dielectric constant is about 12 and the leakage current density is lower than $7.78 \times 10^{-8}$ A/cm$^2$ @ 1 MV/cm. The Zr–AlO$_x$ dielectric film prepared by spin coating method embodies good process and electrical characteristics. It is expected to be applied in the field of printing and flexible electronics.

## 4. Conclusions

In this study, we report a high-$k$ Zr–AlO$_x$ dielectrics were fabricated by sol–gel spin coating at low temperature annealing process. From the results of surface tension measurement, Zr doping can lower the surface tension and the viscosity of precursor solutions, resulting in better wetting. Moreover, the doping of Zr also reduces the internal weak bonds and defects after the films is cured. Thereby, the amorphous films with high quality can be prepared at low temperature. Combined with the main factors carried by minitab analysis and experimental results, we obtained the optimal process with

0.05-mol/L Zr-doping and 200 °C annealing temperature. The relative dielectric constant and leakage current density of the film are, respectively 12 and $7.78 \times 10^{-8}$ A/cm$^2$ @ 1 MV/cm.

**Author Contributions:** Conceptualization, Y.Z.; methodology, Z.L. and S.Z.; software, W.Z.; investigation, Y.Z. and Z.L.; resources, R.Y. and H.N.; writing—original draft preparation, Y.Z.; writing—review & editing, Y.Z., Z.L., and H.N.; visualization, Y.Z. and W.Z.; supervision, R.Y, Y.W., and H.N.; project administration, J.P.; funding acquisition, Z.Z. and T.Q. All authors have read and agreed to the published version of the manuscript.

**Funding:** This work was supported by Key-Area Research and Development Program of Guangdong Province (No.2020B010183002), National Natural Science Foundation of China (Grant Nos. 51771074 and 61574061), the Major Integrated Projects of National Natural Science Foundation of China (Grant No. U1601651), Guangdong Major Project of Basic and Applied Basic Research (Grant No. 2019B030302007), Science and Technology Project of Guangzhou (Grant No. 201904010344) and Fundamental Research Funds for the Central Universities (Grant No. 2019MS012), Ji Hua Laboratory scientific research project (Grant No. X190221TF191).

**Conflicts of Interest:** The authors declare no conflict of interest.

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
