# Peer review of "Effect of Zirconium Doping on Electrical Properties of Aluminum Oxide Dielectric Layer by Spin Coating Method with Low Temperature Preparation"

_coatings, doi:10.3390/coatings10070620_

Round 1

Reviewer 1 Report

First of all, title of the manuscript is very confusing. It is not clear that what authors want to explain by present title. As it is mention in title that there are two types of study: first of Zirconium doping at low temperature and second is electrical properties of Aluminum oxide which was produced by spin coating methods. These are two different study.

However, there is no novelty in present study where same authors have published identical work instead of lower in annealing temperature as: Zirconium-Aluminum-Oxide Dielectric Layer with High Dielectric and Relatively Low Leakage Prepared by Spin-Coating and the Application in Thin-Film Transistor, Coatings 2020, 10, 282; doi:10.3390/coatings10030282. Also authors skip this reference in introduction rather they have used in discussion section. However, some other authors also have used Zr doping to fabricate AlOx gate at: Solution-deposited Zr-doped AlOx gate dielectrics enabling high-performance flexible transparent thin film transistors, https://pubs.rsc.org/en/content/articlepdf/2013/tc/c3tc30550c. This manuscript shows 28% similarity index. On these reasons, I reject the manuscript.

Moreover, it is very important to fix the title then my following suggestion can help authors to improve the quality of the manuscript.

  1. Abstract is not clear. Authors need to rewrite it again.
  2. Whenever authors are writing et al., please do not use first name of authors. I am giving one example: Christophe Avis et al. It should be Avis et al. not Christophe Avis et al.
  3. Throughout the manuscript, English editing is required.
  4. Experimental section is not convincing and details. Authors have not provided the details about the samples to a particular experiment.
  5. Authors have not described the results shown in Fig. 2.
  6. Lines 141-152 are not clear and confusing. Are authors discussing about Fig. 2 or 3? Whatever value and details are discussed in these lines are not correlating with figures as mentioned by authors.
  7. There is no difference in value of capacitance-voltage of samples annealed at 150 and 200 oC as well as concentration of Zr doping as shown in Fig. 5a-b then how authors can mention that as the annealing temperature and concentration of Zr is increased the dielectric constant is decreased and increased, respectively. It needs to be justified or replot the graphs for better clarification.

Overall manuscript is not written scientifically and lack of characterization. Authors need to characterize the film using XRD, FT-IR, SEM and TEM.

Author Response

Dear Reviewer: 
Thank you for your valuable comments concerning our manuscript.  At the request of the reviewer, we change the title into “Effect of Zirconium Doping on Electrical Properties of Aluminum Oxide Dielectric Layer by Spin Coating Method with Low Temperature Preparation”. Those comments are valuable and very helpful for revising and improving our paper. We have studied comments carefully and have made correction which we hope meet with approval. The revision and the manuscript are attached.

Thank you and best regards

Reviewer 2 Report

The manuscript by Yue et al discusses how Zr addition to AlOx can improve the spin-coating proces of the dielectric film depostion, as well as the dielectric properties when annealed at low temperature. High-k dielectrics are crucial to modern electronics, and this research is relevant and adequate for the scope of the journal Coatings.

Whereas the Introduction of the manuscript is well written, the Results section was clearly written by someone with a poor grasp of English, and it needs thorough copy editing. In addition, the results are often discussed in ways unbecoming of a research journal and read more like a lab report of a PhD student, discussing for example how least squares fitting was perfomed in Origin. Equally jarring is Eq.4 of the capacitance of a thin film, at Grade School physics level, even detailing the value of vacuum permittivity. 

The chemical insights on why the spin coating process and the annealing temperature may change with Zr inclusion, as well as the effects on dielectric properties, or the putative double layer capacitance, are all interspersed into the poorly written memory-dump-style Research section, These should be moved to a Discussion section, as well as thoroughly rewriting the Research section.

The 3D plots of Fig 3 and 5 use a color scale that reaches into unphysical negative values, this should be corrected.

The Figure caption of Fig. 5 is inadequate, each panel must be described, the labels of panel d are not legible.

Fig. 5 seems to imply that the dielectric constant is the same for any concentration and for the lower annealing temperatures. How can Fig. 7 then show important dependencies?

The I-V data of Fig. 8 does not seem to show any consistent trends with concentration, or even with annealing. How do the dependecies in Fig. 9 come about?

In any event, since Fig. 7 seems to be data extracted from Fig. 5, and in the same manner Fig. 9 seems to be data extracted from Fig. 8, these would be better combined into only two figures: 3+5, and 8+9, with Fig. 6 moved to the Discussion.

The higher dielectric contant is obtained at the lowest annealing temperature in this work. But would such an 'unannealed' film be technologically useful otherwise? This is not discussed, even though the stated aim of the study is to decrease the annealing temperature and increase the dielectric constant.

Author Response

Dear Reviewer: 
Thank you for your valuable comments concerning our manuscript.  At the request of the reviewer, we change the title into “Effect of Zirconium Doping on Electrical Properties of Aluminum Oxide Dielectric Layer by Spin Coating Method with Low Temperature Preparation”. Those comments are valuable and very helpful for revising and improving our paper. We have studied comments carefully and have made correction which we hope meet with approval. The revision and the manuscript are attached.

Thank you and best regards.

Reviewer 3 Report

The manuscript "Effect of Zirconium Doping on Low Temperature Preparation and Electrical Properties of Aluminum Oxide Dielectric Layer by Spin Coating Method" by Zhou Yue, Liang Zhihao, Yao Rihui, Zuo Wencai, Zhou Shangxiong, Zhu Zhennan, Wang Yiping, Qiu Tian, Ning Honglong, and Peng Junbiao describes how the doping of AlOx solutions with Zr can improve the deposition of Zr-AlOx films via spin coating. This topic is certainly of interest, as it explores new methods to enable low-cost, solution-processed production of electronic devices. However, the following points need to be addressed, before publication may happen:

1. In the introduction, the authors should also mention other on-going strategies to improve the performance and affordability of dielectrics, for example such as polymer dielectrics that can be also spin coated (https://doi.org/10.1021/acsami.8b20827) or efforts to improve performance by patterning techniques (https://doi.org/10.1117/12.2500286).

2. In line 88, the authors explain how to obtain a suitable thickness of the layers (via repeating the process three times). Can they authors comment on what effect lower spin speeds and/or different concentrations would have on the film thickness? Would different spin speeds or concentrations be detrimental to the film properties? If so, could the authors elaborate this point further?

3. How many samples were measured in Figure 7? This question arises, because in this plot, the e_r value of the middle concentration looks like a potential outlier (if only one sample was tested or one measurement was performed). Also, in this plot the concentrations listed on the x-axis look mismatched with the actual data points.

4. As an additional analysis, the authors could calculate the differential resistance R_diff of the tested MIM devices (see Figure S5a of the following paper: https://doi.org/10.1021/acs.jpcc.8b10730). This would enable them to determine something like a shunt resistance, which would be a reasonable way to compare the propensity of these films with regards to leakage.

5. The authors mentioned the bandgaps of Al and Zr. Did the authors perform any absorption and/or transmission measurements of these films? This could be quite interesting in the scope of transparent or semi-transparent electronics.

6. The authors should use contour map plots rather then 3D plots in Fig 3f and 5d. The former are easier to discern and better to obtain the actual data, whereas 3D plots might look fancier, but due to the perspectives are quite poor to extract the actual data.

7. The authors should double check the spelling in the paper to improve the quality of the manuscript further. Some errors were highlighted in the attached document, but these are by no means all mistakes.

Author Response

(The authors gave the same response as above.)

Round 2

Reviewer 1 Report

authors have improved the manuscript and answered all question scientifically.

Author Response

Dear Reviewer: 

Thank you again for your recognition of our manuscript. Those comments are valuable and very helpful for improving the quality of the paper. 

Best regards,

Zhou Yue 

Reviewer 3 Report

The revised manuscript "Effect of Zirconium Doping on Electrical Properties of Aluminum Oxide Dielectric Layer by Spin Coating Method with Low Temperature Preparation" by Zhou Yue, Liang Zhihao, Yao Rihui, Zuo Wencai, Zhou Shangxiong, Zhu Zhennan, Wang Yiping, Qiu Tian, Ning Honglong, and Peng Junbiao describes how the doping of AlOx solutions with Zr can improve the deposition of Zr-AlOx films via spin coating. This topic is certainly of interest, as it explores new methods to enable low-cost, solution-processed production of electronic devices. The authors of the paper have addressed most of the points and concerns brought forth by the reviewers. However, some smaller issues still remain:

1. In Figure 9d, the authors should double check the legend of the plot, since it does not match with what was described in the text (see line 241 and following). The text describes the 0.05 M Zr doped device annealed at 200 °C as the device with the highest shunt resistance. However, in the plot in Fig 9d, this is not the case, rather the 0.075 M Zr at 200 °C has the highest shunt resistance, at least according to the legend. Therefore, I ask the authors to take extra care to make sure that the values in this figure are correctly labelled as well as to double check the other parts of the manuscript to avoid these types of mistakes.

2. There are some minor layout and style issues (e.g. the caption of Fig. 1 being on a different page than the actual figure, bracketed citations being in front of the periods of sentences etc.) as well as some spelling and typing errors (missing periods etc.). As mentioned, these are minor issues, but equally, this means they can be easily corrected and will improve the (perceived) quality of the manuscript even further.

Author Response

Dear Reviewer: 

Thank you for your comments concerning our manuscript. Those comments are valuable and very helpful for improving the quality of the manuscript. We have marked the revision in blue in the paper. We hope it can meet with your approval. The Reply Letter is attached.

Thank you and best regards.

Zhou Yue
